# Exploring lumbar and lower limb kinematics and kinetics for evidence that lifting technique is associated with LBP

Nic Saraceni[1]*, Amity Campbell[1], Peter Kent[1,2], Leo Ng[1], Leon Straker[1], Peter O'Sullivan[1,3]

1 Curtin School of Allied Health, Curtin University, Perth, Western Australia, Australia, 2 Department of Sports Science and Clinical Biomechanics, University of Southern Denmark, Odense, Denmark, 3 Body Logic Physiotherapy Clinic, Shenton Park, Western Australia, Australia

☯ These authors contributed equally to this work.
* Nic.Saraceni@curtin.edu.au

**Data Availability Statement:** All relevant data are within the paper and its Supporting Information files.

## Abstract

### Purpose

To investigate if lumbar and lower limb kinematics or kinetics are different between groups with and without a history of LBP during lifting. Secondly, to investigate relationships between biomechanical variables and pain ramp during repeated lifting.

### Methods

21 LBP and 20 noLBP participants completed a 100-lift task, where lumbar and lower limb kinematics and kinetics were measured during lifting, with a simultaneous report of LBP intensity every 10 lifts. Lifts were performed in a laboratory setting, limiting ecological validity.

### Results

The LBP group used a different lifting technique to the noLBP group at the beginning of the task (slower and more squat-like). Kinetic differences at the beginning included less peak lumbar external anterior shear force and greater peak knee power demonstrated by the LBP group. However, at the end of the task, both groups lifted with a much more similar technique that could be classified as more stoop-like and faster. Peak knee power remained greater in the LBP group throughout and was the only kinetic difference between groups at the end of the lifting task. While both groups lifted using a more comparable technique at the end, the LBP group still demonstrated a tendency to perform a slower and more squat-like lift throughout the task. Only one of 21 variables (pelvic tilt at box lift-off), was associated with pain ramp in the LBP group. Conclusions: Workers with a history of LBP, lift with a style that is slower and more squat-like than workers without any history of LBP. Common assumptions that LBP is associated with lumbar kinematics or kinetics such as greater lumbar flexion or greater forces were not observed in this study, raising questions about the current paradigm around 'safe lifting'.

**Funding:** An Australian Government Research Training Program Scholarship was received by the lead author to support his capacity to undertake this research. A third party physiotherapy organisation (Biosymm) provided a $50 voucher for the participant's time spent in data collection. Neither funder played any role in the design, analysis, or reporting of this study.

**Competing interests:** The authors have no competing interests to declare.

## Introduction

Low back pain (LBP) is the world's most disabling condition and leading cause of work absenteeism [1]. Lifting is a common risk factor for LBP development and engaging in manual tasks that are heavy or in awkward postures has been reported to increase the risk of LBP persistence [2–4]. Attempts to reduce the risk of LBP related to lifting in the workplace have included both task redesign and worker training in safe lifting techniques [5–9]. The guiding principles of these interventions aim to reduce lumbar forces and exposure to lumbar flexion, especially when they are combined. These lifting risk reduction strategies have been largely extrapolated from historical cadaveric studies which identified porcine spinal segments to be less tolerant to compression when in a more flexed position [10–13]. This has led to the view that LBP or injury related to lifting is caused by incorrect lifting. Stoop lifting, which involves a more horizontal thorax, greater kyphotic (flexed) curvature of the lumbar spine and straighter knees is considered to increase the susceptibility of the lumbar tissues to load and strain [14]. In contrast, squat lifting, which involves a more vertical thorax, less kyphotic (flexed) lumbar spine and greater knee bend is therefore thought to be optimal and reduce the likelihood of tissue strain and injury [14–16]. Based on these assumptions, manual handling training commonly advocates squat lifting and warns against stoop lifting to reduce the risk of LBP [17].

However, a recent systematic review found there was no longitudinal or cross-sectional in-vivo evidence that people with LBP flex the lumbar spine more during lifting than those without, casting doubt on this long-held assumption [18]. The existing evidence was of low quality. Furthermore, Nolan et al [19] conducted a systematic review that investigated full body kinematics in people with and without LBP during lifting. In contrast to popular belief, the review concluded that people with LBP lifted with a more vertical thorax and deeper knee bend (more a squat-like lift), as well as slower and with less spinal range of movement (ROM), compared to pain-free people who lifted with a faster and more stoop-style lift. However, the authors highlighted that the current evidence had numerous limitations, including a lack of reporting of LBP intensity (2 of 9 studies) and disability levels (4 of 9 studies). This review also did not investigate forces on the lumbar spine.

When measuring kinematics of the whole body during lifting, both the positioning and velocity of each body segment affects the distribution and magnitude of forces on lumbar spine structures [20]. Given the close relationship between kinematics and kinetics, studies that have investigated both are particularly useful. The few studies that have compared lumbar forces and kinematics between people with and without LBP during natural lifting report mixed findings. Studies by Marras et al [21,22], found people with LBP had greater net external moment and internal compressive force during lifting. This was due to the people with LBP lifting with less trunk flexion (increasing the moment arm from spine to the load) and greater trunk muscle co-contraction, which resulted in greater internal lumbar compression forces than the group without LBP. In contrast, work by Lariviere et al [23], found no differences in forces on the lumbar spine in people with and without LBP during lifting. Participants in the Marras et al study, had greater pain (5.4/10) compared to (2.7/10) those in the Lariviere et al study, which may have influenced the way participants lifted. There is a surprising paucity of studies investigating forces on the lumbar spine in people with and without LBP, given the common messaging to 'squat with a straight back to reduce load', when lifting in occupational health settings. Further, no study reporting lumbar biomechanics in groups of people with LBP during lifting, has investigated if any of these biomechanical factors are associated with a change in pain intensity during repeated lifting (termed 'pain ramp' in this study). The only two studies investigating pain ramp during lifting, have explored relationships between non-biomechanical factors (quantitative sensory testing and psychological factors) and pain ramp

[24,25]. Manual handling advisors and others involved in healthcare, commonly advocate more squat-like lifting, even though there is no support from in-vivo research that the biomechanics of lifting and LBP intensity during lifting are associated [17,26,27].

There are a number of other limitations of the existing evidence that hamper the ability to inform people about 'safe' lifting technique. For example, previous studies have not recruited manual workers both with and without a history of LBP. Interestingly, no study to our knowledge has specifically recruited manual workers who have engaged in repetitive lifting for many years, without reporting LBP. Given prospective studies are difficult to conduct, these workers may hold clues to how people should lift in order to reduce the risk of LBP. It is also unknown from previous studies if the participants recruited to LBP groups actually experienced LBP that was provoked by or related to lifting [19]. Furthermore, a number of studies captured less than 10 lifts, of one weighted object (pen to 11.4kg box) and only during symmetrical lifts from directly in front [18]. Many of these lifting studies did not induce fatigue and the task was not pain provocative, potentially limiting the ecological validity of those findings. No study has previously reported power generated at the lumbar spine, hips and knees during lifting in manual workers, so it is unknown if lifting with greater knee bend is of any mechanical benefit (i.e. reduced lumbar power). Lastly, there are no studies that have reported whole body kinematics and lumbar kinetics over the duration of a repeated lifting task simultaneously with change in LBP during lifting. So, it is not known if relationships exist between biomechanical factors and potentially escalating LBP intensity during a repeated lifting task.

Therefore, the aims of this study were to investigate during a repetitive lifting task that replicated work demands:

1. Whether there are differences in lumbar and lower limb kinematics or kinetics of manual workers with and without a history of lifting-related LBP, and

2. In those workers with a history of lifting-related LBP, whether lumbar and lower limb kinematics or kinetics are associated with change in LBP intensity over the repeated lifting task.

## Method

### Participants

Participants (28 males and 14 females) volunteered for this study and were recruited from workplaces (e.g. trades, shelf stackers, stock picker and packers) through phone calls, flyers and emails. Participants were recruited to either a LBP group (n = 21) or a noLBP group (n = 21) matched for similar age, height and weight. Males and females were recruited, which was assisted by a third-party physiotherapy organisation (Biosymm Physiotherapy, Perth, Australia) who also reimbursed participants $50AUD for their time. Ethical approval was granted from the Human Research Ethics Committee at Curtin University (HRE2018-0197) and written informed consent was obtained. The individual in this manuscript has given written informed consent (as outlined in PLOS consent form) to be photographed for Fig 1.

Manual workers with and without a history of LBP were recruited. All of the included participants must have been >18 years of age, currently working in manual jobs >20 hours per week and involved in regular lifting (>25 lifts/shift).

**The LBP group also satisfied the following additional criteria.**   Dominant axial LBP (between T12 and gluteal fold) that was chronic or recurrent for greater than 3 months duration; lifting must have been a primary aggravating factor (repeated lifting at work increased low back pain to a level of >3/10). All of the above had to have been met and also one of the following two criteria (i) at least 1 episode in the past 12 months where they were unable to

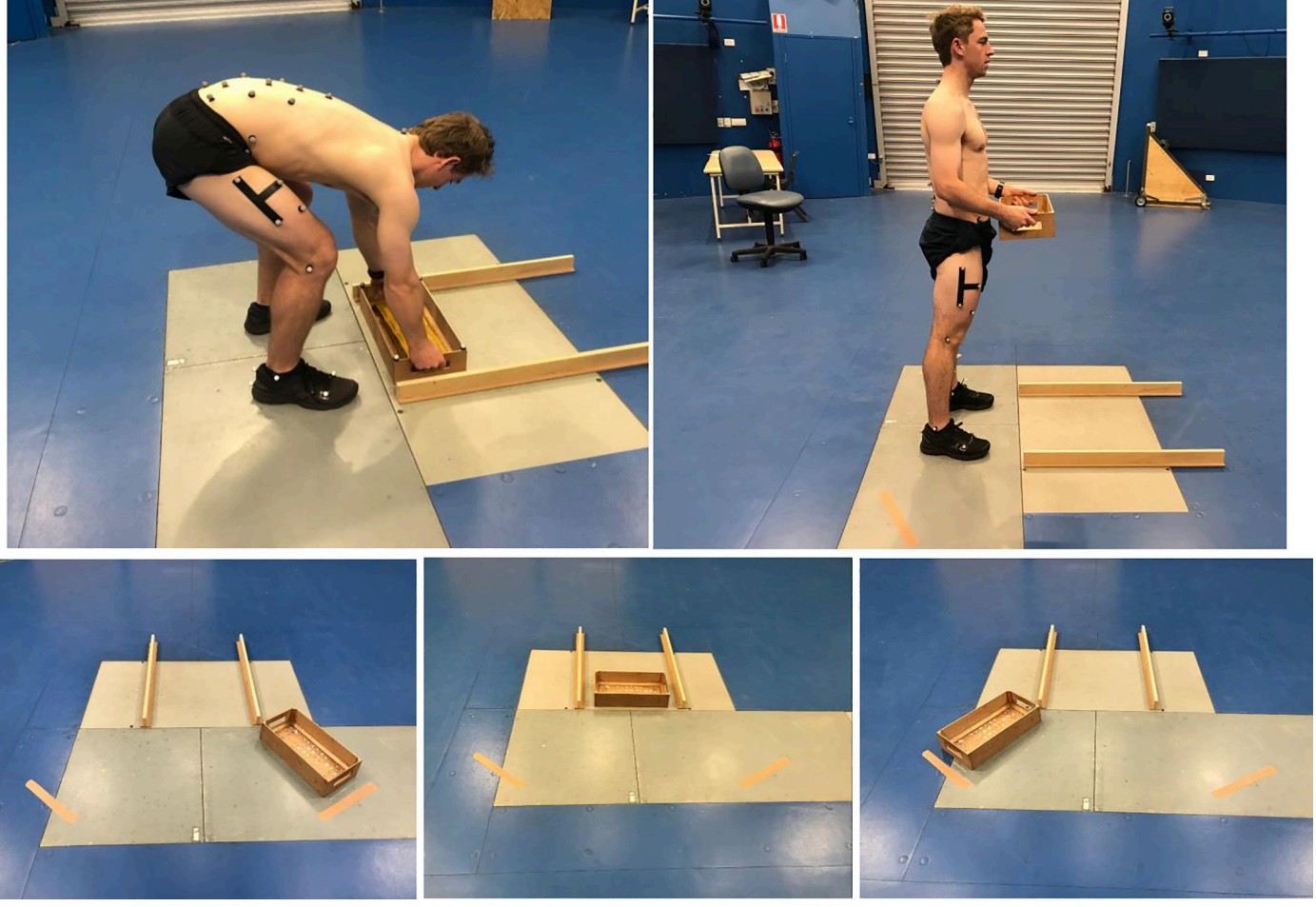

**Fig 1. Lifting task images.** The upper two images demonstrate a symmetrical lift. The three lower images are examples of all lift type origins used in the task for symmetrical and asymmetrical lifts.

attend work or they had to modify how or what they lift at work because of LBP or have taken medication for LBP or have seen a health practitioner for LBP. And (ii) average weekly low back pain (past week) $\geq$ 3/10. Regular exclusion criteria applied for biomechanical LBP studies such as acute lumbar radiculopathy.

**The noLBP group satisfied the following additional criteria.** No history of disabling LBP over the past 5 years. This meant participants had never missed a day of work or made any change in activity levels due to LBP, had no LBP exceeding 24 hours that was greater than 3/10 intensity on a numerical pain rating scale (NPRS) and had not seen a healthcare worker for LBP [28].

## Experimental design

**Lifting procedure.** Data for this laboratory study were collected during a 2-hour measurement session for each participant in 2019. The lifting task comprised 25 lifts (5 symmetrical and 20 asymmetrical) with an empty box (200 grams), followed by 75 lifts (15 symmetrical and 60 asymmetrical) with a box mass set at 10% of each participant's body mass. All lifts were from the floor and participants were encouraged to perform the task in whichever way they felt they normally would, to reflect how they naturally lift at work. There was no set cadence

for the lifting task. A detailed description of the testing procedures is provided in S1 File and demonstrated in Fig 1.

Upon arrival at the Curtin University motion analysis laboratory, retro-reflective markers were placed on specific anatomical landmarks in accordance with gold standard trunk and lower limb 3D motion analysis methods [29,30]. Subject-specific static calibration trials were conducted with markers placed on the medial and lateral malleoli and medial and lateral femoral condyles in addition to the markers described in S1 Table [31]. An 18-camera VICON MX motion analysis system (Vicon, Oxford Metrics, Oxford, UK) operating at 250 Hz and three 1.2 m × 1.2 m force plates (Advanced Mechanical Technology Inc., Watertown, MA) sampling at 2000 Hz were used to collect kinematic and ground reaction force data.

**Variables.** Participant characteristics were collected, including age (years), biological sex (male/female), height (m) and mass (kilograms). Prior to the lifting task, ratings of average pain over the past week and current pain pre-lifting task were also collected (NPRS 0–10). Both the feeling of fatigue (0–10 modified Borg Scale) and LBP intensity (NPRS 0–10) were measured following every 10 lifts while participants continued lifting [32]. Dependent variables included lumbar and lower limb kinematic and kinetic variables that were calculated during lifting and lowering phases of each lift, as defined in S2 Table.

Lumbar kinematic variables included peak intra-lumbar and lumbo-pelvic flexion, lateral flexion and rotation. Peak: thorax inclination, hip flexion, knee flexion, ankle dorsiflexion and heel height, as well as pelvic tilt at box lift off, were also calculated. Peak and average lumbo-pelvic and thorax velocities were calculated using the central difference method.

Lumbar kinetic variables included three-dimensional peak: power, net moment and external forces acting on the spine at the L5/S1 joint. Peak lumbar forces were separated into compression, lateral shear and anterior shear. Peak power was also calculated at the hip and knee. All kinetics were normalized to body mass. Scaled inertial parameters for the lower limb [33], pelvis and lumbar segments [34] were incorporated in the inverse dynamics model for the calculation of lumbo-pelvic kinetics. Further details of the biomechanical modelling are provided in S2 File.

**Data processing.** The 3D data were processed using Vicon Nexus motion analysis software (Vicon, Oxford Metrics, Oxford, UK). Data were filtered using a fourth-order low-pass Butterworth filter operating at a cut-off frequency of 2 Hz, as determined using a residual analysis.

A lift was defined from the initiation of trunk forward bending (without the box) to the end of trunk extension following the box being placed at its destination location (using a combination of the angular velocity of the C7 marker, movement of the intra-lumbar spine and box position in a customised Labview program (National Instruments, Austin, Tx, USA)). In order to examine repeated lifting data, each lift was time normalised from initiation of trunk forward bending (0%) to the end of trunk extension (100%).

All data were inspected for outliers (i.e. >2SD from the mean) and where present, that specific lift was further visualised in Vicon. Where possible, a reason for any outlier was recorded (e.g. box not placed on force plate) and that particular outlier datum was removed from analysis. Each lift had two phases: a lifting phase (bend without box and lift with box) and a lowering phase (lowering with box and return to start position without box). The change point from lifting to lowering was when the trunk changed direction from rising to lowering (defined by the C7 marker angular velocity and box movement). Kinematic and kinetic variables are reported for each phase separately (Table 2 - lifting phase and S3 Table– lowering phase). The hip, knee and ankle variables were collected bilaterally and averaged. The left and right side peak kinematics and kinetics were all significantly correlated. The reporting of the average of both sides did not alter the results of the paper and therefore was used to improve the readability.

**Sample size.** Based on a pre-hoc estimation of 100 lifts per participant, the Optimal Design Plus software (hlmsoft.net) was used to calculate the target sample size. A sample of 40 participants (20 LBP, 20 noLBP) was estimated to conservatively provide 80% power to detect a 2.5˚ (6%) difference in peak absolute intra-lumbar flexion (an effect size of 0.30, based on the Sanchez-Zuriaga 2011 data) [35], with a p-value of 0.05, given a peak absolute lumbar flexion angle of 41.1˚ (SD 8.6˚).

**Data analysis.** Between-group comparisons of demographic, pain and fatigue variables were analysed with bias-corrected bootstrapped (100 samples with replacement) linear regression. Between-group comparisons of kinematic and kinetic variables used multi-level linear mixed models, bias-corrected bootstrapped with the unadjusted and adjusted estimates reported [36,37]. In the unadjusted models, the dependent variable was the kinematic or kinetic variable of interest, the level 1 independent variables were group (LBP/NoLBP), time (lift number) and the interaction between them, and the level 2 variables were participant ID and lift number. The adjusted models also controlled for age, sex, height and weight, lift type (symmetrical/asymmetrical), box weight (unloaded/loaded), the group by lift type interaction, and the box weight by lift type interaction. Those two interactions were retained because they were statistically significant (p<0.05) during preliminary model building.

To identify whether kinematics or kinetics were associated with change in LBP intensity (pain ramp), over a repeated lifting task, we also used bias-corrected bootstrapped multi-level linear mixed models, where pain ramp was the dependant variable and a single kinematic or kinetic variable was the independent variable. Pain ramp was defined as the change in pain from lift 0 to lift 100 in each participant. Only kinematics or kinetics that were different between groups were tested in this way, and were analysed in unadjusted (a single independent variable) and adjusted (the addition of age and sex) forms. All statistical analyses were performed using STATA version 15.1 (StataCorp, College Station, TX, USA) with a p-value of <0.05 as the threshold for statistical significance.

One female in the noLBP group was removed from analysis as her biomechanical data was not considered sufficiently accurate due to marker placement and movement related to high adiposity. Three further male participants' kinetic data were removed (two LBP and one noLBP) due to an error made calibrating the laboratory or incorrect marker placement during data collection.

## Results

The data of 41 participants were used for the kinematic analyses and 38 participants were used for kinetic analyses. The demographic variables of the participants were similar between groups **Table 1**. All participants completed the 100 lift task.

**Table 1. Characteristics of worker participants both with a history of low back pain (LBP) and without a history of low back pain (noLBP).**

|  | LBP (n = 21) | noLBP (n = 20) | Between Group Differences |
|---|---|---|---|
| Demographics |  |  |  |
| Sex, Female (%) | 7/21 (33.3%) | 6/20 (30%) | 3.3% (2.3–4.3%) |
| Age (years) | 37.7 (31.1–44.2) | 32.5 (27.6–37.4) | 5.2 (-2.2–12.6 |
| BMI (kg/m2) | 24.0 (23.0–25.1) | 24.0 (22.7–25.4) | -0.0 (-1.6–1.6) |
| Pain and fatigue |  |  |  |
| Pain–Average previous week (0–10 Scale) | 3.5 (2.7–4.3) | 0.3 (0.0–0.6) | **3.1 (2.2–4.0)** |
| Pain–Entering lab (0–10 Scale) | 1.9 (1.3–2.6) | 0.4 (0.1–0.7) | **1.5 (0.8–2.2)** |
| Pain–Beginning of lifting task (0–10 Scale) | 1.6 (1.0–2.3) | 0.1 (-0.0–0.3) | **1.5 (0.8–2.2)** |
| Pain–End of lifting task (0–10 Scale) | 3.8 (2.7–4.9) | 0.8 (0.3–1.3) | **3.0 (1.8–4.1)** |
| Fatigue–post lifting task (Borg Perceived Exertion Scale 0–10) | 6.5 (5.5–7.4) | 3.3 (2.0–4.6) | **3.2 (1.6–4.8)** |

Between-group comparisons of demographic, pain and fatigue variables were analysed with bias-corrected bootstrapped (100 samples with replacement) linear regression. Data are mean and (95%CI) unless otherwise stated.

### Aim 1- are there differences in how people with and without LBP lift?

During the lifting phase, compared to manual workers with no history of LBP, the LBP group began the repeated lifting task using a technique that was more squat-like (Table 2). That is, they lifted with a deeper knee bend (26.4˚ greater), more vertically inclined thorax (15.3˚) and pelvis (16.6˚), less intra-lumbar flexion (-4.9˚) and greater ankle dorsiflexion (9.8˚). Peak thorax (-21.8˚/sec) and lumbar (-18.3˚/sec) bending velocities and also average thorax (-16.0˚/sec) and lumbar (-14.9˚/sec) velocities were slower in the LBP group. Only two kinetic measures were different, with the LBP group demonstrating greater peak knee power (0.2W/kg) and less peak external lumbar anterior shear force (-0.7 N/kg) at the beginning of the repeated lifting task.

Of all the differences that were observed during the first lifts of the repeated task between groups, only higher peak knee power and a slower lumbar peak bending velocity persisted in the LBP group when performing the final lifts. Therefore, the groups were much more similar in lifting kinematics and kinetics at the end of the task compared to the beginning. Fig 2 depicts the lifting kinematics of the LBP and noLBP groups at the beginning and completion of the lifting task, with greater squat in the LBP group at the beginning, followed by a transition to a more stoop-like style over time. The results for the lowering phase (loaded bending and unloaded return), were similar to that which occurred in the lifting phase (unloaded lowering and loaded return), except for peak intra-lumbar flexion and thorax inclination, which remained less flexed in the LBP group throughout the task (further detailed in S3 Table).

### Aim 2—are kinematics or kinetics associated with change in LBP intensity (pain ramp) over a repeated lift task in the LBP group?

Pain intensity in the LBP group increased on average by 2.2 points on a 0 to 10 point NPRS over the duration of the 100 lift task, although, individual pain responses varied (Fig 3).

Of the 21 biomechanical variables that were different when comparing between groups, during either the lifting or lowering phase, only pelvic position at box lift off, during the lifting phase was associated with pain ramp in the LBP group (adjusted coefficient 0.008 (95%CI 0.000 to 0.017, P = 0.042)) (Tables 3 and S4).

## Discussion

### Summary of results

This cross-sectional study addressed a number of limitations of the in-vivo lifting-related LBP literature. For the first time, lumbar and lower limb kinematics and kinetics of long term (>5 years) manual workers who have not experienced disabling LBP, were compared to a group of manual workers with a history of disabling LBP related to, and aggravated by, lifting. Further, we deem the criteria of this lifting task an important distinguishing feature of our study. Ground level asymmetrical lifts are where risk of injury is reported to be greatest and often have not been included in previous research [38–40]. Most occupational lifting is not from an optimal position directly in front of the body [41], and the repeated 100 lift task allowed for the analysis of biomechanical changes over time between groups. At the start of the repeated lifting task the group with LBP demonstrated less peak intra-lumbar spine flexion, greater vertical orientation of the thorax and pelvis and greater peak knee flexion (more squat-like lift),

**Table 2. Comparison of kinematics and kinetics during lifting phase for workers with (LBP) and without (noLBP) a history of lifting related low back pain.**

| | | Group values (95%CI) (unadjusted) | | Difference (unadjusted) | Difference (adjusted*) |
|---|---|---|---|---|---|
| SPATIAL KINEMATICS | | | | | |
| | | **LBP** | **noLBP** | | |
| Peak intra-lumbar flexion†‡ | **Lift 1** | **14.4˚ (12.3 to 16.6)** | **19.6˚ (17.6 to 21.6)** | **-5.2˚ (-8.1 to -2.3) P<0.001** | **-4.9˚ (-8.0 to -1.8) P = 0.002** |
| | Lift 95 | 19.2˚ (17.0 to 21.5) | 21.8˚ (19.7 to 23.8) | -2.6˚ (-5.5 to 0.3) P = 0.084 | -2.8˚ (-6.3 to 0.6) P = 0.105 |
| Peak lumbo-pelvic flexion | Lift 1 | 16.5˚ (13.9 to 19.2) | 16.4˚ (14.0 to 18.9) | 0.1˚ (-3.5 to 3.7) P = 0.963 | 0.6˚ (-3.2 to 4.4) P = 0.768 |
| | Lift 95 | 17.3˚ (14.9 to 19.7) | 15.5˚ (11.2 to 19.8) | 1.8˚ (-3.4 to 7.0) P = 0.500 | 2.4˚ (-2.6 to 7.4) P = 0.355 |
| Peak intra-lumbar lateral flexion | Lift 1 | 5.7˚ (4.9 to 6.5) | 5.2˚ (4.3 to 6.1) | 0.5˚ (-0.7 to 1.7) P = 0.430 | 0.2˚ (-1.2 to 1.6) P = 0.776 |
| | Lift 95 | 6.0˚ (5.1 to 6.9) | 5.4˚ (4.7 to 6.1) | 0.6˚ (-0.6 to 1.8) P = 0.321 | 0.5˚ (-0.9 to 1.9) P = 0.454 |
| Peak lumbo-pelvic lateral flexion†‡ | Lift 1 | 3.4˚ (2.9 to 3.8) | 3.4˚ (2.8 to 4.0) | 0.0˚ (-0.7 to 0.7) P = 0.959 | 0.0˚ (-0.7 to 0.8) P = 0.903 |
| | Lift 95 | 3.4˚ (3.0 to 3.8) | 4.0˚ (3.3 to 4.7) | -0.5˚ (-1.4 to 0.3) P = 0.181 | -0.8˚ (-1.9 to 0.4) P = 0.192 |
| Peak intra-lumbar rotation‡ | Lift 1 | 2.8˚ (2.3 to 3.2) | 2.6˚ (2.3 to 2.9) | 0.2˚ (-0.3 to 0.7) P = 0.479 | 0.1˚ (-0.5 to 0.7) P = 0.794 |
| | Lift 95 | 3.0˚ (2.4 to 3.5) | 2.9˚ (2.6 to 3.2) | 0.1˚ (-0.6 to 0.8) P = 0.820 | 0.0˚ (-0.8 to 0.8) P = 0.982 |
| Peak lumbo-pelvic rotation | Lift 1 | 3.1˚ (2.5 to 3.6) | 3.0˚ (2.5 to 3.5) | 0.0˚ (-0.7 to 0.8) P = 0.895 | 0.1˚ (-0.9 to 1.1) P = 0.809 |
| | Lift 95 | 3.4˚ (2.8 to 4.0) | 3.1˚ (2.6 to 3.6) | 0.3˚ (-0.4 to 1.0) P = 0.397 | 0.5˚ (-0.4 to 1.4) P = 0.278 |
| Peak thoracic inclination†‡ | **Lift 1** | **81.3˚ (77.0 to 85.6)** | **94.5˚ (87.9 to 101.0)** | **-13.2˚ (-21.4 to -5.0) P = 0.002** | **-15.3˚ (-25.1 to -5.4) P = 0.002** |
| | Lift 95 | 99.6˚ (95.4 to 103.7) | 102.8˚ (97.3 to 108.3) | -3.2˚ (-10.1 to 3.6) P = 0.353 | -7.0˚ (-14.9 to 0.9) P = 0.082 |
| Pelvic inclination at box lift off† | **Lift 1** | **36.9˚ (32.3 to 41.6)** | **50.2˚ (43.7 to 56.6)** | **-13.2˚ (-21.2 to -5.2) p = 0.001** | **-16.6˚ (-26.0 to -7.1) p = 0.001** |
| | Lift 95 | 47.5˚ (42.2 to 52.9) | 51.3˚ (45.0 to 57.6) | -3.8˚ (-12.3 to 4.7) P = 0.386 | -6.1˚ (-15.3 to 3.1) P = 0.196 |
| Peak hip flexion | Lift 1 | 107.9˚ (105.4 to 110.5) | 108.0˚ (105.1 to 110.9) | -0.1˚ (-3.7 to 3.6) P = 0.974 | 1.0˚ (-3.4 to 5.4) P = 0.652 |
| | Lift 95 | 107.9˚ (105.4 to 110.5) | 109.1˚ (104.2 to 114.1) | -1.2˚ (-6.8 to 4.4) P = 0.671 | -0.4˚ (-6.1 to 5.4) P = 0.900 |
| Peak knee flexion | **Lift 1** | **115.1˚ (106.1 to 124.0)** | **91.5˚ (79.0 to 104.0)** | **23.6˚ (7.8 to 39.4) P = 0.003** | **26.4˚ (8.1 to 44.7) P = 0.005** |
| | Lift 95 | 95.5˚ (85.4 to 105.6) | 85.9˚ (73.2 to 98.6) | 9.6˚ (-5.5 to 24.6) P = 0.211 | 12.8˚ (-3.9 to 29.4) P = 0.132 |
| Peak ankle dorsiflexion† | **Lift 1** | **34.7˚ (32.0 to 37.5)** | **25.2˚ (21.3 to 29.2)** | **9.5˚ (4.5 to 14.5) P<0.001** | **9.8˚ (4.3 to 15.2) P<0.001** |
| | Lift 95 | 26.7˚ (23.3 to 30.1) | 22.7˚ (18.6 to 26.8) | 4.0˚ (-1.0 to 9.0) P = 0.119 | 4.8˚ (-0.2 to 9.8) P = 0.061 |
| Peak heel lift (mm) | **Lift 1** | **59.3 (49.9 to 68.7)** | **44.1 (40.8 to 47.4)** | **15.2 (5.7 to 24.8) p = 0.002** | **14.2 (1.5 to 26.8) P = 0.028** |
| | Lift 95 | 57.4 (49.1 to 65.7) | 46.5 (43.1 to 50.0) | **10.9 (1.9 to 19.8) P = 0.018** | 8.2 (-2.8 to 19.2) P = 0.143 |
| TEMPORAL KINEMATICS | | | | | |
| Peak lumbar (L1-L5) segment velocity relative to the vertical (deg/s)‡ | **Lift 1** | **86.0 (78.6 to 93.3)** | **99.7 (91.3 to 108.0)** | **-13.7 (-25.0 to -2.4) P = 0.018** | **-18.3 (-29.5 to -7.0) P = 0.001** |
| | **Lift 95** | **102.5 (96.2 to 108.7)** | **114.4 (106.9 to 121.9)** | **-11.9 (-21.6 to -2.0) P = 0.017** | **-13.1 (-25.2 to -1.1) P = 0.033** |

*(Continued)*

**Table 2.** (*Continued*)

| | | Group values (95%CI) (unadjusted) | | Difference (unadjusted) | Difference (adjusted*) |
|---|---|---|---|---|---|
| Average bending lumbar (L1-L5) segment velocity relative to the vertical (deg/s)‡ | **Lift 1** | **39.6 (34.6 to 44.6)** | **52.1 (46.6 to 57.6)** | **-12.5 (-20.3 to -4.7) P = 0.002** | **-14.9 (-22.9 to -6.8) P<0.001** |
| | Lift 95 | 49.3 (44.5 to 54.2) | 57.8 (51.6 to 64.1) | **-8.5 (-16.6 to -0.3) P = 0.041** | -8.8 (-18.3 to 0.6) P = 0.067 |
| Average lifting lumbar (L1-L5) segment velocity relative to the vertical (deg/s)†‡ | Lift 1 | -27.8 (-32.8 to -22.8) | -30.9 (-35.7 to -26.1) | 3.0 (-4.2 to 10.3) P = 0.411 | 3.0 (-5.9 to 12.0) P = 0.503 |
| | Lift 95 | -26.4 (-34.2 to -18.6) | -20.4 (-27.7 to -13.2) | -5.9 (-16.3 to 4.4) P = 0.261 | -2.8 (-13.2 to 7.6) P = 0.599 |
| Peak Thorax velocity (C7-T10 segment inclination deg/s)‡ | **Lift 1** | **105.0 (96.6 to 113.4)** | **119.7 (109.9 to 129.4)** | **-14.7 (-28.3 to -1.1) P = 0.033** | **-21.8 (-35.7 to -8.0) P = 0.002** |
| | Lift 95 | 129.4 (120.8 to 138.0) | 138.3 (126.3 to 150.4) | -8.9 (-23.6 to 5.7) P = 0.232 | -10.9 (-28.8 to 7.1) P = 0.235 |
| Average Thorax velocity (C7-T10) segment inclination (bend with no box deg/s) | **Lift 1** | **50.7 (45.7 to 55.7)** | **63.2 (57.2 to 69.2)** | **-12.5 (-20.8 to -4.9) P = 0.003** | **-16.0 (-24.9 to -7.4) P<0.001** |
| | Lift 95 | 63.6 (58.0 to 69.2) | 70.7 (62.6 to 78.8) | -7.1 (-17.2 to 2.9) P = 0.162 | -7.9 (-19.4 to 3.6) P = 0.179 |
| Average Thorax velocity (C7-T10 segment inclination (lift with box deg/s) †‡ | Lift 1 | -37.7 (-43.4 to -32.0) | -38.8 (-44.0 to -32.0) | 1.1 (-7.0 to 9.1) P = 0.795 | 0.5 (-9.3 to 10.4) P = 0.913 |
| | Lift 95 | -37.1 (-45.8 to -28.4) | -26.7 (-35.3 to -18.2) | -10.4 (-22.4 to 1.7) P = 0.091 | -5.4 (-17.2 to 6.3) P = 0.365 |
| KINETICS | | | | | |
| Peak lumbar power (Normalised to body mass W/kg)‡ | Lift 1 | 0.8 (0.7 to 0.9) | 0.8 (0.7 to 0.9) | -0.0 (-0.2 to 0.2) P = 0.999 | 0.0 (-0.1 to 0.2) P = 0.913 |
| | Lift 95 | 0.9 (0.8 to 1.0) | 1.0 (0.9 to 1.1) | -0.1 (-0.2 to 0.0) P = 0.141 | -0.0 (-0.1 to 0.1) P = 0.758 |
| Average lumbar power (Normalised to body mass W/kg) (bend with no box)‡ | Lift 1 | 0.4 (0.3 to 0.4) | 0.4 (0.3 to 0.4) | -0.0 (-0.1 to 0.0) P = 0.464 | -0.0 (-0.1 to 0.1) P = 0.634 |
| | Lift 95 | 0.4 (0.3 to 0.4) | 0.4 (0.4 to 0.5) | -0.0 (-0.1 to 0.0) P = 0.206 | -0.0 (-0.1 to 0.0) P = 0.479 |
| Average lumbar power (Normalised to body mass W/kg) (lift with box)‡ | Lift 1 | -0.3 (-0.3 to -0.2) | -0.3 (-0.3 to -0.2) | -0.0 (-0.1 to 0.0) P = 0.589 | -0.0 (-0.1 to 0.1) P = 0.895 |
| | Lift 95 | -0.3 (-0. to -0.2) | -0.2 (-0.3 to -0.1) | -0.0 (-0.1 to -0.0) P = 0.257 | -0.0 (-0.1 to 0.0) P = 0.209 |
| Peak hip power (Normalised to body mass W/kg)† | Lift 1 | 1.4 (1.1 to 1.6) | 1.4 (1.3 to 1.6) | -0.1 (-0.4 to 0.2) P = 0.539 | -0.1 (-0.4 to 0.3) P = 0.668 |
| | Lift 95 | 1.6 (1.2 to 2.0) | 1.3 (1.0 to 1.5) | 0.4 (-0.1 to 0.9) P = 0.126 | 0.2 (-0.1 to 0.6) P = 0.221 |
| Peak knee power (Normalised to body mass W/kg)‡ | **Lift 1** | **1.1 (0.9 to 1.2)** | **0.7 (0.5 to 0.9)** | **0.3 (0.1 to 0.6) P = 0.004** | **0.2 (0.0 to 0.5) P = 0.023** |
| | **Lift 95** | **0.8 (0.6 to 1.0)** | **0.6 (0.4 to 0.7)** | **0.3 (0.0 to 0.5) P = 0.045** | **0.3 (0.0 to 0.7) P = 0.028** |
| Peak lumbar moment (Normalised to body mass) (NM/kg)‡ | Lift 1 | 2.4 (2.3 to 2.5) | 2.4 (2.2 to 2.5) | 0.0 (-0.1 to 0.2) P = 0.920 | 0.0 (-0.2 to 0.2) P = 0.996 |
| | Lift 95 | 2.7 (2.6 to 2.9) | 2.8 (2.6 to 3.0) | -0.1 (-0.3 to 0.2) P = 0.583 | 0.0 (-0.3 to 0.2) P = 0.690 |
| Peak lumbar external anterior shear force (Normalised to body mass) (N/kg)‡ | **Lift 1** | **4.5 (4.2 to 4.8)** | **4.9 (4.6 to 5.2)** | **-0.4 (-0.8 to 0.0) P = 0.032** | **-0.7 (-1.2 to -0.1) P = 0.010** |
| | Lift 95 | 5.8 (5.4 to 6.1) | 6.1 (5.8 to 6.1) | -0.3 (-0.8 to 0.1) P = 0.106 | -0.3 (-0.8 to 0.1) P = 0.103 |

(*Continued*)

**Table 2.** (Continued)

| | | Group values (95%CI) (unadjusted) | | Difference (unadjusted) | Difference (adjusted*) |
|---|---|---|---|---|---|
| Peak lumbar lateral shear force (Normalised to body mass) (N/kg) †‡ | Lift 1 | 0.5 (0.4 to 0.5) | 0.5 (0.4 to 0.6) | 0.0 (-0.1 to 0.0) P = 0.320 | -0.1 (-0.2 to 0.1) P = 0.239 |
| | Lift 95 | 0.6 (0.6 to 0.7) | 0.8 (0.7 to 0.9) | -0.1 (-0.2 to 0.0) P = 0.054 | -0.1 (-0.3 to 0.0) P = 0.175 |
| Peak lumbar external compression force (Normalised to body mass) (N/kg)‡ | Lift 1 | 4.2 (4.1 to 4.3) | 4.3 (3.9 to 4.7) | -0.1 (-0.5 to 0.3) P = 0.681 | 0.0 (-0.5 to 0.5) P = 0.917 |
| | Lift 95 | 4.8 (4.6 to 5.0) | 4.9 (4.5 to 5.3) | -0.1 (-0.6 to 0.4) P = 0.697 | -0.2 (-0.9 to 0.4) P = 0.472 |

*Adjusted for age, sex, height, weight, an interaction between pain group and lift type, an interaction between box weight and lift type and lift number.

† Significant interaction between group and time.

‡ Significant time effect.

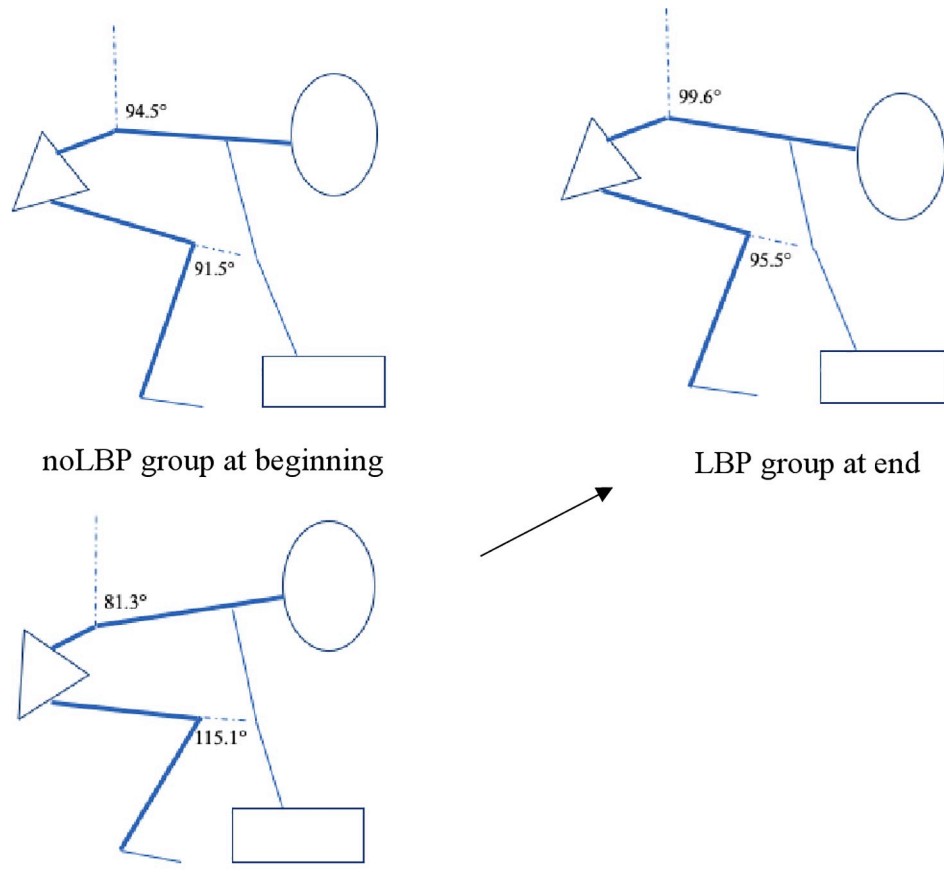

**Fig 2. Representation of change in kinematics over time in the LBP group.** Peak knee flexion and peak thorax inclination angles during the lifting phase are highlighted in the images. By the end of the lift trial, the LBP group transition toward a more stoop-like style, similar to that of the noLBP group.

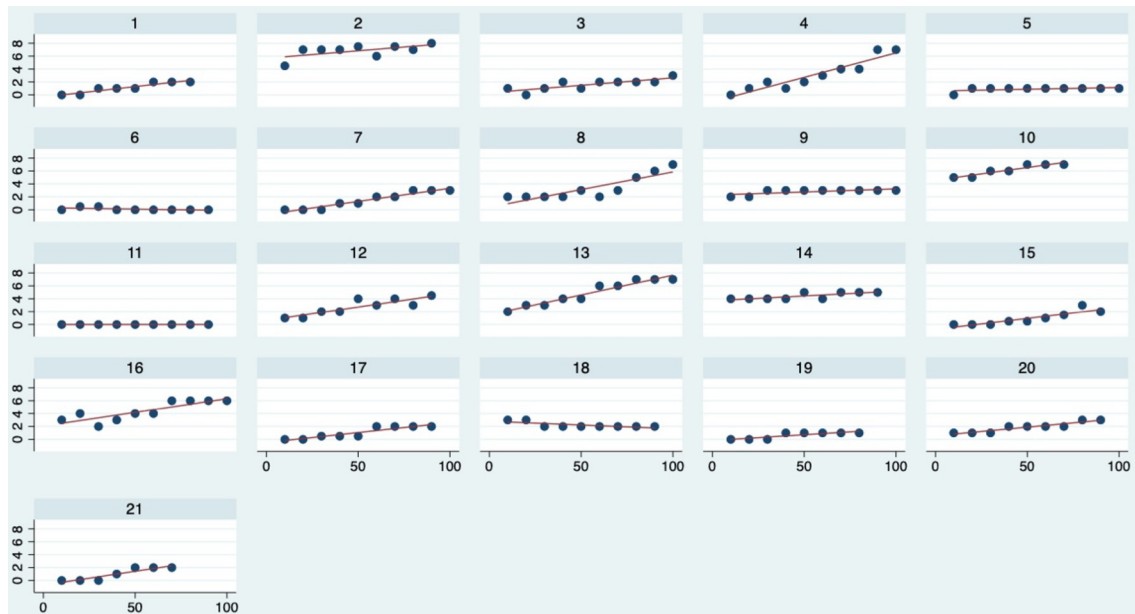

**Fig 3. Pain ramp in the LBP group by participant.** The y axis represents LBP intensity on a numerical pain rating score (0–10) which was asked after every 10 lifts. The x axis represents lift number. Not all 100 lifts were usable for data analysis for every participant based on various errors in data capture.

compared to the group without LBP. The group with LBP also moved slower in general, across both the lifting and lowering phases, and had less estimated peak external lumbar anterior shear force. By the end of this 100-lift task, while both groups lifted using a more comparable

**Table 3. In those workers with a history of low back pain (LBP group), associations between pain ramp during the lifting phase and each kinematic or kinetic variable that was different between groups.**

|  | Unadjusted Coefficient (95% CI) | p-value | Adjusted* Coefficient (95% CI) | p-value |
|---|---|---|---|---|
| *Lifting Phase—Spatial kinematics* |  |  |  |  |
| Peak intra-lumbar flexion | 0.010 (-0.009 to 0.029) | 0.320 | 0.009 (-0.012 to 0.029) | 0.370 |
| Peak thorax inclination (C7-T10 segment inclination relative to the vertical) | 0.005 (-0.003 to 0.012) | 0.210 | 0.005 (-0.003 to 0.012) | 0.213 |
| Peak knee flexion | -0.004 (-0.008 to 0.001) | 0.111 | -0.004 (-0.008 to 0.001) | 0.109 |
| Peak ankle dorsiflexion | **-0.014 (-0.028 to -0.000)** | **0.049** | -0.014 (-0.028 to 0.000) | 0.053 |
| Peak heel lift | 0.002 (-0.003 to 0.008) | 0.435 | 0.002 (-0.003 to 0.008) | 0.415 |
| **Pelvic inclination at box lift off** | **0.009 (0.000 to 0.017)** | **0.042** | **0.008 (0.000 to 0.017)** | **0.042** |
| *Lifting phase—Temporal kinematics* |  |  |  |  |
| Peak lumbar velocity (L1-L5 segment inclination relative to the vertical) | 0.002 (-0.004 to 0.008) | 0.508 | 0.002 (-0.004 to 0.007) | 0.530 |
| Average lumbar velocity during unloaded bending phase (L1-L5 segment inclination relative to the vertical) | 0.003 (-0.004 to 0.010) | 0.390 | 0.003 (-0.004 to 0.010) | 0.408 |
| Peak thorax velocity (C7-T10 segment inclination relative to the vertical) | 0.001 (-0.003 to 0.005) | 0.587 | 0.001 (-0.003 to 0.005) | 0.617 |
| Average thorax velocity during unloaded bending phase (C7-T10 segment inclination relative to the vertical) | 0.002 (-0.003 to 0.008) | 0.394 | 0.002 (-0.003 to 0.008) | 0.416 |
| *Lifting phase—kinetics* |  |  |  |  |
| Peak knee power | -0.005 (-0.131 to 0.122) | 0.941 | -0.007 (-0.134 to 0.120) | 0.913 |
| Peak external anterior shear force | 0.064 (-0.102 to 0.230) | 0.452 | 0.062 (-0.105 to 0.228) | 0.468 |

* Adjusted for age and sex.

technique that was more aligned to a stoop-like lift (Fig 2), the LBP group still demonstrated a tendency to perform a slower and more squat-like lift throughout the task. Despite bending their knees more at the beginning and generating higher peak knee power throughout the task, this had no influence on peak lumbar power which was similar between groups for the entirety of the lifting task. Of the 21 variables that were different between groups (which were mostly at the beginning of the task), the only biomechanical variable related to pain ramp was pelvic inclination at box lift off, during the lifting phase. As no other kinetic or kinematic measure was associated with pain ramp, and people with LBP lifted more like the group without LBP over time, this relationship between pelvic position and pain ramp may be a chance finding.

## Aim 1 –comparison of lifting technique between LBP and noLBP groups

Our findings are broadly in line with a recent systematic review that found people with LBP lift with a slower and deeper squat with less spinal ROM when compared to people without pain who lift faster and more stooped [19]. Our study addressed a number of the limitations identified by this systematic review including: poor description of participants and recruitment method, low number of lifts, heterogeneity across included LBP groups and questionable validity of how lumbar spine flexion was measured. This was also the first study to compare the range of biomechanical variables that have previously been thought to expose the lumbar spine to pain or injury during lifting. We included lumbar; posture, forces, velocity and power during lifting, and compared between groups with and without LBP. By providing both lower limb and lumbar kinematics and kinetics over repeated lifts, this study provides novel insight into the relationship between whole body biomechanics and lifting related LBP.

There is a growing body of cross-sectional evidence demonstrating that people with LBP tend to lift and bend in a manner that is different to people without LBP, using a more squat-like technique [19] with less lumbar flexion [18,28]. Interestingly, this lifting technique is advocated in manual handling training to reduce the risk of LBP [9]. Although this cross-sectional study cannot determine the cause-effect relationship of LBP related to lifting, or the reason for the differences seen in lifting technique, previous research has demonstrated that people both with LBP [42,43] and without LBP [44], who have higher levels of pain-related fear, lift with less lumbar flexion. There is also evidence that as people with chronic LBP improve, how they perform activities such as bending and lifting becomes faster with greater lumbar ROM in most cases [45]. Together this work suggests that people with LBP perceive squat-like lifting to be 'protective' during bending and lifting and therefore move accordingly.

A unique finding of our study, is the tendency for people with LBP to shift with repeated lifting over time towards a more stoop-like lifting technique (more horizontal thorax, knee extension etc). Even so, some differences remained at completion of the task, where the LBP group lifted slower with less intra-lumbar flexion. Previous studies that have shown differences between groups with and without LBP, have usually only measured less than 10 lifts in a laboratory environment, which is unlikely to replicate what occurs in a manual workplace [35,46–48]. In our study, the LBP group reported greater levels of fatigue (6.5/10) during the lifting task compared to the group without LBP (3.3/10), which may partly explain this change in lifting technique over the duration of the task. Previous studies report that a stoop style lift is preferred by workers performing frequent heavy manual tasks as this lifting style has been shown to be more energy efficient compared to squat lifting [49–52]. Quadriceps muscle fatigue is thought to be one reason why repetitive squat lifting is not sustainable [51,53]. Therefore, advising people in workplaces to lift with a more vertical inclination of the spine and deeper knee bend is less efficient and may not be realistic for people engaged in repeated lifting. Furthermore, workplace prevention strategies targeting lifting with a 'straight' back have not been

shown to be effective [6–8], although one reason for this lack of efficacy could be that this advice is not being followed.

There is debate as to whether any lifting technique is superior at reducing net forces on the lumbar spine and therefore its application to risk reduction related to LBP [54–57]. In our study, both groups' external lumbar peak; anterior shear forces, net moment and power increased over time and were similar between groups at the end of the task. This has been shown in other studies in repeated and fatiguing lifting and seems a normal response to repetitive lifting [52]. This has not been investigated previously in people with and without LBP. Increases in lumbar kinetics with repeated lifts in our study, are likely due to greater peak velocities and greater peak thorax flexion. While it is believed that higher peak lumbar forces combined with greater lumbar flexion is associated with lifting-related LBP, we did not observe this. The idea that lumbar flexion combined with high lumbar forces causes LBP, has been extrapolated from in-vitro studies [11,12,58], theorised based on spinal modelling [59–61] and inferred from intra-discal pressure studies [62–64]. To date no in-vivo data exists to support this view [18].

## Aim 2 –association of pain ramp in the LBP group and changes in kinetics and kinematics during lifting

To our knowledge, our study is the first to investigate the relationship between changes in lifting biomechanics and changes in pain intensity (pain ramp) over a repeated lifting task. On average there was a 2.2 point increase in pain intensity in the LBP group during the repeated lifting task, which is similar to previous reports of pain ramp related with repeated lifting and bending in people with LBP [25,65]. This affirms that this lifting task was pain provocative for many of these manual workers. While most of the LBP group reported increasing pain intensity over the 100 lifts, 2 of 21 LBP participants did not experience greater pain and 1 of 21 experienced pain reduction following the lifting task (Fig 3). While we specifically recruited people with a history of lifting-related LBP, not all participants demonstrated pain ramp during our lifting tasks, suggestive of variability in responses to lifting and the episodic nature of LBP in a cohort with a known history of lifting related-LBP. These findings are consistent with previous reports of variable pain responses to repeated movement [24,66]. Previous studies investigating pain ramp during repeated bending or repeated lifting have not measured lumbar kinematics or kinetics [24,25].

Only 1 out of 21 variables that were different when comparing lifting biomechanics between groups was associated with pain ramp in the LBP group, suggesting that there was no conclusive evidence that any single biomechanical factor explained pain ramp during repeated lifting. Pelvic tilt at box lift off during the lifting phase, was the only variable associated with pain ramp in the LBP group. This one finding may be spurious, as no other biomechanical variable showed a relation with pain ramp and the two groups became more homogeneous over the duration of the lifting task. Further, a post-hoc sensitivity analysis (S5 Table) was conducted with removal of the three participants who did not experience an increase in pain with repeated lifts. The beta coefficients changed by +/- 0.001 or 0.002, and the only previously significant association (pelvic tilt at box lift off during the lifting phase) was no longer statistically significant. Therefore, based on our results, this sensitivity analysis reinforced the observation that there is no conclusive evidence that single biomechanical factors explain pain responses to repeated lifting.

Historically, singular biomechanical variables, such as lumbar compressive force or lumbar flexion during lifting, were thought to drive lifting related-LBP through tissue injury/strain mechanisms [15,67–70]. In this study, people with and without LBP moved more similarly by the end of the 100 lift task and experienced similar peak lumbar forces yet had very different pain experiences.

When interpreting the findings of this study, it could be argued that squat lifting represents a protective/helpful strategy for people with LBP and the transition towards more stooped lifting over the 100 lifts did co-occur with increased pain for these people. To be clear, the LBP group were still more squat-like compared to the noLBP group at the end of the task and both groups transitioned to more stoop-like lifting with repeated lifts. However, when we tested the association between biomechanical parameters and pain ramp, no associations were found (except one, which is likely a type 1 error). These findings do not support the contention that squat lifting was a protective/helpful strategy for people with LBP. As there seems no clear biomechanical advantage of more squat-like lifting based on our results or any other research in persistent LBP [21,22], the reason people with LBP adopt a more squat-like lift remains unknown.

The lack of association between lifting biomechanics and pain ramp highlights the potential role of non-biomechanical factors in a person's pain experience [71,72]. For example, previous research has demonstrated a relationship, in people with LBP, between greater pain ramp following a repeated bending and lifting task and higher levels of psychological distress and also greater sensitivity to pressure and cold stimuli [24]. In another study of people with chronic LBP, there was a relationship with greater pain ramp during repeated lifting and higher levels of pain related fear [25]. Given the lack of evidence for biomechanical associations with pain ramp in our study, future research should also investigate the interplay of non-biomechanical factors on a person's pain experience during lifting.

## Strengths and limitations

This study is the first to explore the biomechanics of repeated lifting in manual workers with lifting-related LBP compared to a group of manual workers engaged in repeated lifting for over 5 years with no history of LBP. The lifting task involved two different load weights, symmetrical and asymmetrical lifts, men and women, and was pain provoking for those with LBP. This allowed participants to move in a natural unconstrained manner for 100 lifts and attempted to replicate similar demands of workplace lifting occupations with gold standard recording of kinematics and kinetics during lifting. Pain intensity was tracked after every 10 lifts which allowed for the exploration of relationships between lifting technique and pain intensity. The limitations of this study include the use of a lab-based design so there is uncertain ecological validity; people with lifting-related LBP who have not continued manual work were not included in this sample (a potential survivor-bias); and how accurately skin markers represent underlying joint movement is of some debate but regularly performed in this field of research. Further, there was no use of EMG in this study and therefore no internal force estimates were possible. As stated previously, the cross-sectional design precludes insight into cause and effect, so it is unclear whether between-group differences were contributed to by pain or were a response to the pain experience, or neither. The weights lifted were of a low magnitude and therefore we do not know if these results generalise to heavy lifting.

## Implications

There are strong beliefs in both occupational health and clinical settings regarding the importance of squat lifting and minimising lumbar flexion, both for the prevention and management of LBP [7–9,14,17,26,27]. This is commonly taught in the workplace and advocated in clinical and rehabilitation settings [6]. However, there is currently no in-vivo evidence to support this view [18,19,21–23]. In contrast, there is growing evidence that people with LBP demonstrate a lifting technique that they perceive to be 'protective' compared to people without LBP [19,21,22]. That is, people with LBP tend to lift with a deeper knee bend, more vertically

inclined thorax and less intra-lumbar flexion (squat lift) and slower. Ironically, survivors in our study (those who have been employed in manual work for greater than 5 years without LBP) tended to lift in a manner that is considered to increase their risk for the development of LBP (more stoop-like with greater lumbar flexion and faster). This finding is consistent with reports of manual workers who find stoop lifting is more efficient [49,51].

While the cross-sectional nature of this study limits any conclusions regarding causation, the findings raise some interesting questions for both occupational health advisers and clinicians. Given there is currently no clear in-vivo evidence regarding the safest way to lift, it may be more helpful to focus less on the exact lifting technique and more about a person's general health, strength and fitness, lifting efficiency and confidence to engage in repeated lifting tasks. Indeed, telling a person to adopt repeated squat lifting across a day may be fatiguing and unrealistic. This notion is consistent with previous research that has reported the risk of LBP when lifting is greater when a person is tired or fatigued [3]. High quality prospective research into the relationship between lifting biomechanics and the development of LBP is required to better understand risk.

For clinicians managing people with LBP related to lifting, advising them to adopt more squat-like lifting techniques in order to reduce the risk of LBP or manage it lacks evidence. Recent research supports that recovery from chronic and disabling LBP is associated with movement patterns that society perceives to be less safe (a more flexed lumbar spine and faster movement speed) [45]. The lack of clear associations with lifting biomechanics and pain intensity during repeated lifting in this study, suggests a range of factors across biopsychosocial domains should be explored to best assist those with lifting-related LBP [73,74]. Simply focusing on the biomechanics of what has been previously proposed as 'ideal' lifting, may be misguided [75].

## Conclusion

Manual workers with a history of LBP used a slower deeper squat pattern compared to people without LBP at the beginning of a lifting task. Manual workers with >5 years' experience, without a history of LBP, utilized a more stoop-like lifting strategy that was faster. The LBP group transitioned to more stoop-like lifting over the 100 lifts to be more like the noLBP group, although some indicators of what society perceives a 'safer' lifting pattern remained. Pain ramp occurred in the LBP group, although that response was variable across individuals, and there was no conclusive evidence for an association between biomechanical variables and pain ramp. Common assumptions that LBP is associated with lumbar kinematics or kinetics such as greater lumbar flexion or greater forces were not observed in this study, raising questions about the current paradigm around 'safe lifting'.

## Supporting information

**S1 Table. Marker locations.**
(DOCX)

**S2 Table. Description of angular kinematics.**
(DOCX)

**S3 Table. Kinematic and kinetic between group comparisons during lowering phase.**
(DOCX)

**S4 Table. Associations between pain ramp during the lowering phase and each kinematic or kinetic variable that was different between groups—associations for the LBP group**

**only.**
(DOCX)

**S5 Table. In those workers with a history of low back pain (LBP group), associations between pain ramp during the lifting and lowering phases and each kinematic or kinetic variable that was different between groups.** For this sensitivity analysis, those participants (6, 11 and 18) who did not experience pain increase with lifting were removed.
(DOCX)

**S1 File. Lift testing procedures.**
(DOCX)

**S2 File. Biomechanical modelling.**
(DOCX)

# Acknowledgments

The authors acknowledge Professor Anne Smith, Mr Jarrad Kerron and Mr Paul Davey for their contributions to this paper.

# Author Contributions

**Conceptualization:** Nic Saraceni, Leo Ng, Leon Straker, Peter O'Sullivan.

**Data curation:** Nic Saraceni, Peter Kent, Peter O'Sullivan.

**Formal analysis:** Nic Saraceni, Peter Kent, Leon Straker, Peter O'Sullivan.

**Funding acquisition:** Nic Saraceni, Peter O'Sullivan.

**Investigation:** Nic Saraceni, Amity Campbell, Leo Ng, Leon Straker, Peter O'Sullivan.

**Methodology:** Nic Saraceni, Amity Campbell, Peter Kent, Leo Ng, Leon Straker, Peter O'Sullivan.

**Project administration:** Nic Saraceni, Leon Straker, Peter O'Sullivan.

**Resources:** Peter Kent, Peter O'Sullivan.

**Software:** Nic Saraceni.

**Supervision:** Amity Campbell, Peter Kent, Leo Ng, Leon Straker, Peter O'Sullivan.

**Visualization:** Nic Saraceni.

**Writing – original draft:** Nic Saraceni, Amity Campbell, Peter Kent, Leo Ng, Leon Straker, Peter O'Sullivan.

**Writing – review & editing:** Nic Saraceni, Amity Campbell, Peter Kent, Leo Ng, Leon Straker, Peter O'Sullivan.

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
