## [Decision Letter · Decision Letter 0]

28 Apr 2021

PONE-D-21-06065

Exploring lumbar and lower limb kinematics and kinetics for evidence that lifting technique is associated with LBP.

PLOS ONE

Dear Dr. Saraceni,

Thank you for submitting your manuscript to PLOS ONE. After careful consideration, we feel that it has merit but does not fully meet PLOS ONE’s publication criteria as it currently stands. Therefore, we invite you to submit a revised version of the manuscript that addresses the points raised during the review process.

Please, state the criteria for lifting task and associated biomechanical parameters selection. Please, state in the abstract the limted validity of this lifting task to only abstract readers.

We look forward to receiving your revised manuscript.

Kind regards,

Daniel Boullosa

Academic Editor

PLOS ONE

Journal Requirements:

Please include captions for your Supporting Information files at the end of your manuscript, and update any in-text citations to match accordingly. Please see our Supporting Information guidelines for more information: http://journals.plos.org/plosone/s/supporting-information.

3. We note that Figure 1 includes an image of a participant in the study.

Reviewers' comments:

Reviewer's Responses to Questions

**Comments to the Author**

1. Is the manuscript technically sound, and do the data support the conclusions?

Reviewer #1: Yes

Reviewer #2: Yes

2. Has the statistical analysis been performed appropriately and rigorously? 

Reviewer #1: I Don't Know

Reviewer #2: Yes

3. Have the authors made all data underlying the findings in their manuscript fully available?

Reviewer #1: No

Reviewer #2: Yes

4. Is the manuscript presented in an intelligible fashion and written in standard English?

Reviewer #1: Yes

Reviewer #2: Yes

5. Review Comments to the Author

Reviewer #1: General comments

I was highly impressed with many aspects of the study described in this manuscript as well as the manner in which the manuscript was written and presented. Specifically, the rationale for the study was well presented, along with the limitations of the previous literature which then directly informed the design of the study. The selection of the two experimental groups, including the inclusion and exclusion criteria, the number and variety of actual lifting tasks included in the experimental design and the wide variety of kinematic and kinetic measures were also strengths of the study. The provided appendices also provide excellent additional detail regarding aspects of the methods. Some minor ways in which this manuscript can be improved are described in in specific comments section.

Specific comments

Line 74 – 78: could this be a little bit of the chicken and egg type scenario? Is it possible that individuals with LBP may use more of a squat style then stoop style lifting approach as squat style lifting elicits less pain? Thus, is it potentially the only motor control strategy they have available that protects them from increased pain? Such a view may also need to be included within the discussion in some places when interpreting results. Based on your results it may then be when fatigue (especially quadricep and perhaps cardiovascular) increases, they are then forced into using a more stoop lifting pattern which then contributes to increased pain.

Line 138 – 140: was there also any inclusion or exclusion criteria relating to number of years they have been involved in manual lifting in their employment situation? If not, was such data collected from each participant?

Line 160 – 164: while I was impressed with the larger number and variety of lift types that your personal performed competitor other studies, can you provide some context to the potential representative of the number of lifts, height of lift and mass of loads to common lifting requirements in the workplace? Further, where the participants able to lift the 100 loads at a self-selected time and cadence? If so, could such data be presented for each group?

Line 187 – 189: were all these variables collected on just one or two sides of the body? If they were collected on both sides of the body, have you reported an average of both sides in your results?

Line 226 – 236: as the statistical modelling approach described in this section is somewhat complex as a result of your research design, it might be useful to provide some key references that support your statistical approach.

Line 239: while you have used the phrase “pain ramp” many times in the manuscript including the abstract, this appears to be the first time that you attempt to define this variable. As this appears to be a key outcome in your study, I would suggest you need to describe it in some detail within your introduction and provide a more explicit definition and references to support its use in your study.

Line 241 – 243: I understand the rationale for using the kinematic or kinetic variables that differ between groups, but as a result of the large number of dependent variables in the study and the relative strong correlations expected between many of these variables, would a statistical approach to reduce the number of dependent variables such as a PCA have been better to utilise in this case?

Line 244-245: following on from my previous comment regarding a large number of dependent variables, you have conducted a very large of statistical analysis and therefore is it appropriate that a p value of 0.05 be used for every statistical comparison?

Table 1: can you please be more explicit whether these values for the two groups are means and the 95% confidence intervals, as you have used in other tables?

Table 2: are all these comparisons for lift 1 and lift 95? If so, would it be better to compare the first five verse last five lifts to get a better representation of the initial and later lifts of the 100 lift series? Further, your first subcategory of “Kinematic’ is perhaps not completely accurate as your second subcategory of “Velocity” is also a kinematic variable. Therefore, should your first category be something like “Linear and Angular Displacement”?

Line 376: this should read “increased over time”.

Line 394 – 398: would the results of your analysis differ if you excluded these individuals who didn’t experience the pain ramp? I therefore suggest it might be useful to look further into this inter-individual response in pain ramp.

Line 465: this should read “is tired or fatigued”.

Reviewer #2: Dear editor, thank you for giving me the opportunity to review this interesting study.

I think this study is an important piece of research contributing to increase our knowledge in the low back pain field.

I think the manuscript should be accepted for publication after some minor corrections are addressed. Below you can find my comments:

Line 186: brackets are not necessary for the word “appendix”.

Line 191: two dots after the word “peak”

Line 257: I recommend not to use p-values for baseline characteristics in table 1, as recommended by David Moher et al., in their Guideline for Reporting Health Research. Baseline characteristics is a descriptive analysis not inferential analysis.

Table 1: it is not clear what is depicted in brackets, is it range? If so, the value is mean or median? With such a small sample I recommend reporting median (range).

Table 2: the degree º symbol is missing in some data.

Line 375: I think an apostrophe is missing after “groups”

Lines 376: “increase” should be in past perfect tense: increased

Lines 393 to 401: the finding that some participants didn’t worsen, or one even improved, pain with repeated flexion might be explained by the directional preference concept where in patients with discogenic lumbar pain there is a directional movement (flexion, extension or lateral flexion) that improves their symptoms (Surkitt LD, et al. Phys Ther. 2012).

Line 465: “fatigue” should be in past perfect tense: fatigued.

6. PLOS authors have the option to publish the peer review history of their article (what does this mean?). If published, this will include your full peer review and any attached files.

Reviewer #1: **Yes: **Justin Keogh

Reviewer #2: No

---

## [Author Response · Author response to Decision Letter 0]

4 Jun 2021

PLOS One Manuscript PONE-D-21-06065: Exploring lumbar and lower limb kinematics and kinetics for evidence that lifting technique is associated with LBP.

Authors’ response to review comments and suggestions 

May 2021

Thank you, Associate Editor and reviewers, for your insightful comments on this manuscript. Below, we address these concerns and provide details of how we have amended the manuscript to improve it. 

Academic Editor Daniel Boullosa

Editor comments:

1. Please, state the criteria for lifting task and associated biomechanical parameters selection. 

Authors’ response:

Thank you for highlighting this. The lifting task was designed to address the limitations of previous studies in this area. Our study included:

- Symmetrical and asymmetrical lifts from the floor.

- Two different box weights.

- Repeated lifts to induce fatigue, so responses to fatigue/pain/repetition could be investigated in groups of people with and without LBP.

- We encouraged participants to complete the lifting task in their natural way.

Many previous studies comparing lifting biomechanics in people with and without LBP have used lifting tasks that do not represent occupational lifting. Most have captured 3 lifts of a light object and only included lifts from directly in front of the body. Further, a vast majority of biomechanical lifting research instructs participants to lift in a certain way (stoop or squat) and therefore have not captured normal/natural lifting technique.

We have amended the method to highlight the criteria for the lifting task. Appendix 1 and Figure 1 also provide further clarity on lifting task design and requirements (all revised manuscript text is underlined and italicised in this document).

“Data for this laboratory study were collected during a 2-hour measurement session for each participant in 2019. The lifting task comprised 25 lifts (5 symmetrical and 20 asymmetrical) with an empty box (200 grams), followed by 75 lifts (15 symmetrical and 60 asymmetrical) with a box mass set at 10% of each participant’s body mass. All lifts were from the floor and participants were encouraged to perform the task in whichever way they felt they normally would, to reflect how they naturally lift at work. There was no set cadence for the lifting task. A detailed description of the testing procedures is provided in Appendix 1 and demonstrated in Figure 1.”

The biomechanical parameters selected in this study have mostly been studied previously, but usually with poorer quality measures and with lower quality data capture devices. For example, the lumbar spine has been previously measured as a single segment (from L1 to L5), which does not adequately estimate lumbar spine curvature during lifting. Whereas, we used a gold standard data capture device to measure lumbar kinetics and kinematics which included the estimation of lumbar spine curvature (L1-3 segment relative to L3-L5 segment) during lifting. Two recent systematic reviews highlight the previous low quality research in this area (1, 2), and the need for higher quality studies. The uniqueness of this study is that we captured lumbar forces, posture, velocity and power over a fatiguing lifting task, where participants lifted in their natural way. We also captured lower limb kinetics and kinematics together, and therefore the biomechanics of lifting could be better understood. Previous studies have usually looked at one element of lifting risk, such as lumbar forces or lumbar posture during lifting. No study has thoroughly investigated the range of biomechanical variables thought to be important in lifting-related LBP, with a high quality data capture device in relevant cohorts with and without LBP.

We have made further changes to the discussion to more explicitly state the reasoning behind criteria of the lifting task and also the associated biomechanical parameters selected.

“This cross-sectional study addressed a number of limitations of the in-vivo lifting-related LBP literature. For the first time, lumbar and lower limb kinematics and kinetics of long term (>5 years) manual workers who have not experienced disabling LBP, were compared to a group of manual workers with a history of disabling LBP related to, and aggravated by, lifting. Further, we deem the criteria of this lifting task an important distinguishing feature of our study. Ground level asymmetrical lifts are where risk of injury is reported to be greatest and often have not been included in previous research.[37-39] Most occupational lifting is not from an optimal position directly in front of the body,[40] and the repeated 100 lift task allowed for the analysis of biomechanical changes over time between groups.”

And 

“Our findings are broadly in line with a recent systematic review that found people with LBP lift with a slower and deeper squat with less spinal ROM when compared to people without pain who lift faster and more stooped [19]. Our study addressed a number of the limitations identified by this systematic review including: poor description of participants and recruitment method, low number of lifts, heterogeneity across included LBP groups and questionable validity of how lumbar spine flexion was measured. This was also the first study to compare the range of biomechanical variables that have previously been thought to expose the lumbar spine to pain or injury during lifting. We included lumbar; posture, forces, velocity and power during lifting, and compared between groups with and without LBP. By providing both lower limb and lumbar kinematics and kinetics over repeated lifts, this study provides novel insight into the relationship between whole body biomechanics and lifting related LBP.”

2. Please, state in the abstract the limited validity of this lifting task to only abstract readers. 

Authors’ response:

Thank you for highlighting this oversight. We recognise the importance of clarity for abstract only readers. We therefore have amended the abstract as follows:

Abstract

“Purpose: To investigate if lumbar and lower limb kinematics or kinetics are different between groups with and without a history of LBP during lifting. Secondly, to investigate relationships between biomechanical variables and pain ramp during repeated lifting. Methods: 21 LBP and 20 noLBP participants completed a 100-lift task, where gold-standard kinematics and kinetics were measured during lifting, with a simultaneous report of LBP intensity every 10 lifts. Lifts were performed in a laboratory setting, limiting ecological validity. Results: The LBP group used a different lifting technique to the noLBP group at the beginning of the task (slower and more squat-like). Kinetic differences at the beginning included less peak lumbar external anterior shear force and greater peak knee power demonstrated by the LBP group. However, at the end of the task, both groups lifted with a much more similar technique that could be classified as more stoop-like and faster. Peak knee power remained greater in the LBP group throughout and was the only kinetic difference between groups at the end of the lifting task. While both groups lifted using a more comparable technique at the end, the LBP group still demonstrated a tendency to perform a slower and more squat-like lift throughout the task. Only one of 21 variables (pelvic tilt at box lift-off), was associated with pain ramp in the LBP group. Conclusions: Workers with a history of LBP, lift with a style that is slower and more squat-like than workers without any history of LBP. Common assumptions that LBP is associated with lumbar kinematics or kinetics such as greater lumbar flexion or greater forces were not observed in this study, raising questions about the current paradigm around ‘safe lifting’.”

Reviewer 1

Specific comments:

3. Line 74 – 78: could this be a little bit of the chicken and egg type scenario? Is it possible that individuals with LBP may use more of a squat style then stoop style lifting approach as squat style lifting elicits less pain? Thus, is it potentially the only motor control strategy they have available that protects them from increased pain? Such a view may also need to be included within the discussion in some places when interpreting results. Based on your results it may then be when fatigue (especially quadricep and perhaps cardiovascular) increases, they are then forced into using a more stoop lifting pattern which then contributes to increased pain.

Authors’ response:

Thank you for your thoughtfulness about this particularly important interpretation of our results. We understand the logic of this line of reasoning and we considered this view at length based on our results and therefore we have clarified elements of the discussion in-line with your suggestions and comments:

“When interpreting the findings of this study, it could be argued that squat lifting represents a protective/helpful strategy for people with LBP and the transition towards more stooped lifting over the 100 lifts did co-occur with increased pain for these people. To be clear, the LBP group were still more squat-like compared to the noLBP group at the end of the task and both groups transitioned to more stoop-like lifting with repeated lifts.

However, when we tested the association between biomechanical parameters and pain ramp up, no associations were found (except one, which is likely a type 1 error). These findings do not support the contention that squat lifting was protective/helpful strategy for people with LBP. As there seems no clear biomechanical advantage of squat lifting based on our results or any other research in persistent LBP [21, 22], the reason people with LBP squat to lift remains unknown.”

And

“There is a growing body of cross-sectional evidence demonstrating that people with LBP tend to lift and bend in a manner that is different to people without LBP, using a more squat-like technique [19] with less lumbar flexion [18, 28]. Interestingly, this lifting technique is advocated in manual handling training to reduce the risk of LBP [9]. Although this cross-sectional study cannot determine the cause-effect relationship of LBP related to lifting, or the reason for the differences seen in lifting technique, previous research has demonstrated that people both with LBP [41, 42] and without LBP,[43] who have higher levels of pain-related fear, lift with less lumbar flexion. There is also evidence that as people with chronic LBP improve, how they perform activities such as bending and lifting becomes faster with greater lumbar ROM in most cases [44]. Together this work suggests that people with LBP perceive squat-like lifting to be ‘protective’ during bending and lifting and therefore move accordingly.”

4. Line 138 – 140: was there also any inclusion or exclusion criteria relating to number of years they have been involved in manual lifting in their employment situation? If not, was such data collected from each participant?

Authors’ response:

Thank you for highlighting this unintended oversight. The noLBP group must have been in manual work more than 20hours/week for >5 years with no history of LBP. We did not collect the exact hours/week or years in similar employment for each participant, only if they met these criteria or not. The inclusion criteria has been amended as follows:

“The noLBP Group satisfied the following additional criteria:

No history of disabling LBP over the past 5 years. This meant participants had never missed a day of work or made any change in activity levels due to LBP, had no LBP exceeding 24 hours that was greater than 3/10 intensity on a numerical pain rating scale (NPRS) and had not seen a healthcare worker for LBP [28].”

5. Line 160 – 164: while I was impressed with the larger number and variety of lift types that your personal performed competitor other studies, can you provide some context to the potential representative of the number of lifts, height of lift and mass of loads to common lifting requirements in the workplace? Further, where the participants able to lift the 100 loads at a self-selected time and cadence? If so, could such data be presented for each group?

Authors’ response: 

The lifting task attempted to replicate regular work demands of the participants and capture reported lifting injury risk factors. Manual workers in similar occupations, have been reported to lift on average 933 times daily and most frequently between 1 and 10kg.(3) Occupational lifting where the trunk is flexed >60 degrees and rotated >30 degrees has been shown to increase the risk of LBP.(4) Therefore, we incorporated two different box weights (in a weight range of regular occupational lifting) as well as symmetrical and asymmetrical lift types from the floor. We also believed the large number of lifts in our study, helped in capturing natural lifting technique of participants compared to fewer than 10 in previous similar research. Further, greater risk of lifting injury is reported to occur when fatigue and loads increase.(5-7) We attempted to adequately capture this risk in our study, by incorporating a large number of asymmetrical lifts from the floor which were fatiguing and compared the biomechanics of lifting in this high risk position, between groups with and without LBP. The workers were all in occupations that involved repetitive lifting throughout the day (shelf stackers, labourers, tradesmen) and therefore completing 100 lifts in succession was not unfamiliar. 

While we didn’t measure total lift time, we more specifically output the velocity of each body region during the lifting movement. This more detailed velocity information allowed us to see if either group lifted slower or faster. We output both peak and average velocity for all lifting phases (bending unweighted, rising with box, lowering with box and rising unweighted are all reported in Table 2 and Appendix 5). We deemed that output of velocity to be more accurate as it excludes little breaks that may have occurred during the data collection period (small pauses between lifts etc). A self-selected cadence was preferred as we wanted to capture natural lifting. The lifting task took approximately 10mins for all participants to complete, and it was completed with very few breaks, intentionally, so that we did not influence fatigue or pain report over the course of the lifting task.

6. Line 187 – 189: were all these variables collected on just one or two sides of the body? If they were collected on both sides of the body, have you reported an average of both sides in your results?

Authors’ response:

We collected both sides of the body and reported the average. We decided that this was an appropriate approach as statistically the hip, knee and ankle were all significantly correlated in every lift phase to that of the opposite side (Pearsons correlation co-efficients of >0.3). Further, as the reporting of the average of both sides did not alter the result, we deemed that reporting one result improved the readability of the paper.

The following has been added to the methods section of the paper:

“All data were inspected for outliers (i.e. >2SD from the mean) and where present, that specific lift was further visualised in Vicon. Where possible, a reason for any outlier was recorded (e.g. box not placed on force plate) and that particular outlier datum was removed from analysis. Each lift had two phases: a lifting phase (bend without box and lift with box) and a lowering phase (lowering with box and return to start position without box). The change point from lifting to lowering was when the trunk changed direction from rising to lowering (defined by the C7 marker angular velocity and box movement). Kinematic and kinetic variables are reported for each phase separately (Table 2 - lifting phase and Appendix 5 - lowering phase). The hip, knee and ankle variables were collected bilaterally and averaged. The left and right side peak kinematics and kinetics were all significantly correlated. The reporting of the average of both sides did not alter the results of the paper and therefore was used to improve the readability.”

7. Line 226 – 236: as the statistical modelling approach described in this section is somewhat complex as a result of your research design, it might be useful to provide some key references that support your statistical approach.

Authors’ response:

Thank you for this suggestion. We have now included two key references.

Leeden Rvd, Meijer E, Busing FMTA. Resampling Multilevel Models. In: Leeuw Jd, Meijer E, editors. Handbook of Multilevel Analysis. New York, NY: Springer New York; 2008. p. 401-33.

Harrison XA, Donaldson L, Correa-Cano ME, Evans J, Fisher DN, Goodwin CED, et al. A brief introduction to mixed effects modelling and multi-model inference in ecology. PeerJ. 2018;6:e4794.

8. Line 239: while you have used the phrase “pain ramp” many times in the manuscript including the abstract, this appears to be the first time that you attempt to define this variable. As this appears to be a key outcome in your study, I would suggest you need to describe it in some detail within your introduction and provide a more explicit definition and references to support its use in your study.

Authors’ response:

Thank you for highlighting the lack of clarity of the term ‘pain ramp’. We have added detail to the Introduction and Methods.

We have added the following to the introduction:

“There is a surprising paucity of studies investigating forces on the lumbar spine in people with and without LBP, given the common messaging to ‘squat with a straight back to reduce load’, when lifting in occupational health settings. Further, no study reporting lumbar biomechanics in groups of people with LBP during lifting, has investigated if any of these biomechanical factors are associated with a change in pain intensity during repeated lifting (termed ‘pain ramp’ in this study). The only two studies investigating pain ramp during lifting, have explored relationships between non-biomechanical factors (quantitative sensory testing and psychological factors) and pain ramp[24, 25]. Manual handling advisors and others involved in healthcare, commonly advocate more squat-like lifting, even though there is no support from in-vivo research that the biomechanics of lifting and LBP intensity during lifting are associated [17, 26, 27].

And

“To identify whether kinematics or kinetics were associated with change in LBP intensity over a repeated lift task (pain ramp), we also used bias-corrected bootstrapped multi-level linear mixed models, where pain ramp was the dependant variable and a single kinematic or kinetic variable was the independent variable. Pain ramp was defined as the change in pain from lift 0 to lift 100 in each participant. Only kinematics or kinetics that were different between groups were tested in this way, and were analysed in unadjusted (a single independent variable) and adjusted (the addition of age and sex) forms. All statistical analyses were performed using STATA version 15.1 (StataCorp, College Station, TX, USA) with a p-value of <0.05 as the threshold for statistical significance.”

9. Line 241 – 243: I understand the rationale for using the kinematic or kinetic variables that differ between groups, but as a result of the large number of dependent variables in the study and the relative strong correlations expected between many of these variables, would a statistical approach to reduce the number of dependent variables such as a PCA have been better to utilise in this case?

Authors’ response:

Thanks for this query. Yes, the identification and modelling of latent variables (such as using PCA) would be interesting from a causal modelling perspective. However, in this descriptive study we only modelled ‘measured variables’ for two reasons. Firstly, to meet the need for an investigation of a comprehensive range of biomechanical factors that may be associated with LBP during lifting. And secondly, because keeping these variables in their measured forms is more likely to produce results that are more readily interpretable by readers from the clinical and manual handling communities.

10. Line 244-245: following on from my previous comment regarding a large number of dependent variables, you have conducted a very large of statistical analysis and therefore is it appropriate that a p value of 0.05 be used for every statistical comparison?

Yes, this is a good question. The debate in the research community about p-value correction for multiple testing is nuanced and there are diverse opinions about the appropriateness of such procedures. At its core is a concern about balancing Type 1 and Type 2 errors (about balance between finding differences that are actually due to chance compared with overlooking potentially useful findings) especially in exploratory studies, and about reproducibility. 

One consideration is about circumstances where dependent variables might be highly correlated, as the observation that both outcomes result in significant findings in the same direction may reinforce our confidence in the results. In that circumstance, if we were to apply a Bonferroni-type correction, neither outcome might be significant, which is counterintuitive. 

Another consideration is about what number of statistical tests should inform a p-value correction, with authors such as Matsunaga’s (2007) and Rubin (2017) making the distinction between different tests of the same hypothesis and single tests of different hypotheses.(8, 9) Rubin argues that it is most appropriate to conceptualize combined Type I error rate for multiple tests (familywise error) in relation to the number of different tests that are conducted on the same null hypothesis in the same study. Especially in exploratory studies. In our exploratory study, we are testing a large number of different hypotheses, such as ‘are there between group differences in x dependent variable at lift y?’. While we report the unadjusted and adjusted results for each of those hypotheses, that is not usually considered an indication for p-value correction.

In our study, there are 108 hypotheses tested (54 for each of the lifting and lowering phases) about between group differences, with 30 (28%) having p-values below 0.05. That rate of positive findings in this exploratory study does not appear to suggest that they are only the consequence of Type 1 error (5%). There were also 21 hypotheses tested about associations with pain ramp, with 1 (5%) having a p-value below 0.05 and that value becoming non-significant in the sensitivity analysis, supporting our interpretation that there is no conclusive evidence of any association. 

11. Table 1: can you please be more explicit whether these values for the two groups are means and the 95% confidence intervals, as you have used in other tables?

Authors’ response:

Thank you for highlighting this. Data are mean and (95%CI) unless otherwise stated. This has been added to the footnotes of Table 1.

12. Table 2: are all these comparisons for lift 1 and lift 95? If so, would it be better to compare the first five verse last five lifts to get a better representation of the initial and later lifts of the 100 lift series? Further, your first subcategory of “Kinematic’ is perhaps not completely accurate as your second subcategory of “Velocity” is also a kinematic variable. Therefore, should your first category be something like “Linear and Angular Displacement”?

All kinematic and kinetic data were analysed with multi-level linear mixed models, bias-corrected bootstrapped with both the unadjusted and adjusted estimates reported. With this method, Lift 1 and Lift 95 are group-level estimates based on all of the available data (derived from a group-level intercept and slope), using a model with random intercept and slope (that therefore accommodated individual variation). So, use of the first 5 and last 5 lifts, would likely provide a less precise estimate. The reason for using lift 95, and not lift 100, as that the mean number of analysed lifts was 95 due to some data collection issues (such as marker coming loose, miscounting of lift number etc). We have amended Table 2 and Appendix 5 with the wording Spatial and Temporal kinematics to align with the wording previously in Table 3 and Appendix 6 in line with your suggestion.

13. Line 376: this should read “increased over time”.

Authors’ response:

Thank you for highlighting this, it has now been corrected.

14. Line 394 – 398: would the results of your analysis differ if you excluded these individuals who didn’t experience the pain ramp? I therefore suggest it might be useful to look further into this inter-individual response in pain ramp.

Authors’ response:

Thanks for the suggestion. A sensitivity analysis has now been performed by removing the 3 participants whose pain either decreased during the lifts or was unchanged, and repeating these analyses. Typically, the beta coefficients changed by +/- 0.001 or 0.002, and the only previously significant association was no longer statistically significant (Appendix 7). Therefore, this sensitivity analysis reinforced the observation of no conclusive evidence that any single mechanical factor explained pain responses to repeated lifting. 

The following has now been added to the discussion:

“Only 1 out of 21 variables that were different when comparing lifting mechanics between groups was associated with pain ramp in the LBP group, suggesting there was little evidence that any single mechanical factor explained pain responses to repeated lifting. Pelvic tilt at box lift off was the only variable associated with pain ramp in the LBP group. This one finding may be spurious, as no other mechanical variable showed a relation with pain ramp and the two groups became more homogeneous over the duration of the lifting task. Further, a post-hoc sensitivity analysis (Appendix 7) was conducted with removal of the three participants who did not experience an increase in pain with repeated lifts. The beta coefficients changed by +/- 0.001 or 0.002, and the only previously significant association (pelvic tilt at box lift off during the lifting phase) was no longer statistically significant. Therefore, based on our results, this sensitivity analysis reinforced the observation that there is no conclusive evidence that single biomechanical factors explain pain responses to repeated lifting.” 

15. Line 465: this should read “is tired or fatigued”.

Authors’ response:

Thank you for highlighting, this has now been corrected.

Reviewer 2 

16. Line 186: brackets are not necessary for the word “appendix”.

Authors’ response:

Thank you for highlighting, this has now been corrected.

17. Line 191: two dots after the word “peak”

Authors’ response:

Thank you for highlighting, this has now been corrected.

18. Line 257: I recommend not to use p-values for baseline characteristics in table 1, as recommended by David Moher et al., in their Guideline for Reporting Health Research. Baseline characteristics is a descriptive analysis not inferential analysis.

We recognise that reporting guidelines, such as the CONSORT and STROBE Statements discourage reporting statistical tests of baseline differences between groups in randomised controlled trials. We had thought it was useful in cross-sectional case/control studies, to illustrate whether the selection had been successful. We have now added group differences and their 95%CI instead of P values to Table 1. 

19. Table 1: it is not clear what is depicted in brackets, is it range? If so, the value is mean or median? With such a small sample I recommend reporting median (range).

Authors’ response:

Thank you for highlighting this. Between-group comparisons of demographic, pain and fatigue variables were analysed with bias-corrected bootstrapped (100 samples with replacement) linear regression. Therefore, data are mean and (95%CI) unless otherwise stated. This has been added to the footnotes of Table 1.

20. Table 2: the degree º symbol is missing in some data.

Authors’ response:

Thank you for highlighting, this has now been corrected.

21. Line 375: I think an apostrophe is missing after “groups”

Authors’ response:

Thank you for highlighting, this has now been corrected.

22. Lines 376: “increase” should be in past perfect tense: increased

Authors’ response:

Thank you for highlighting, this has now been corrected.

23. Lines 393 to 401: the finding that some participants didn’t worsen, or one even improved, pain with repeated flexion might be explained by the directional preference concept where in patients with discogenic lumbar pain there is a directional movement (flexion, extension or lateral flexion) that improves their symptoms (Surkitt LD, et al. Phys Ther. 2012).

Authors’ response:

Yes we agree that there may be a directional preference component to this person who improved. There is also a myriad of other potential reasons, including that the participant experiences a latent onset of pain (i.e. they benefit from repeated movement at the time, but after ceasing the task experience an increase in pain in the subsequent hours/days). As we did not assess the source of pain (disc or other) nor tested pain responses in the hours/days following the task, we would rather not speculate on why some people increased in pain and others did not. We do think this is an interesting area of future research to be investigated with a different study design. We do acknowledge that our findings are in-line with previous research suggesting people with LBP experience different pain responses to repeated movement and therefore we have added the reference you have suggested as well as made a change to the discussion: 

“While we specifically recruited people with a history of lifting-related LBP, not all participants demonstrated pain ramp during our lifting tasks, suggestive of variability in responses to lifting and the episodic nature of LBP in a cohort with a known history of lifting related-LBP. These findings are consistent with previous reports of variable pain responses to repeated movement [24, 67]. Previous studies investigating pain ramp during repeated bending or repeated lifting have not measured lumbar kinematics or kinetics [24, 25].

24. Line 465: “fatigue” should be in past perfect tense: fatigued.

Authors’ response:

Thank you for highlighting, this has now been corrected. 

References

1. Nolan D, O'Sullivan K, Newton C, Singh G, Smith BE. Are there differences in lifting technique between those with and without low back pain? A systematic review. Scand J Pain. 2020;20(2):215-27.

2. Saraceni N, Kent P, Ng L, Campbell A, Straker L, O'Sullivan P. To Flex or Not to Flex? Is There a Relationship Between Lumbar Spine Flexion During Lifting and Low Back Pain? A Systematic Review With Meta-analysis. The Journal of orthopaedic and sports physical therapy. 2020;50(3):121-30.

3. Jakobsen MD, Sundstrup E, Brandt M, Persson R, Andersen LL. Estimation of physical workload of the low-back based on exposure variation analysis during a full working day among male blue-collar workers. Cross-sectional workplace study. Applied ergonomics. 2018;70:127-33.

4. Hoogendoorn WE, Bongers PM, de Vet HC, Douwes M, Koes BW, Miedema MC, et al. Flexion and rotation of the trunk and lifting at work are risk factors for low back pain: results of a prospective cohort study. Spine. 2000;25(23):3087-92.

5. Steffens D, Ferreira ML, Latimer J, Ferreira PH, Koes BW, Blyth F, et al. What triggers an episode of acute low back pain? A case-crossover study. Arthritis care & research. 2015;67(3):403-10.

6. Coenen P, Kingma I, Boot C, Twisk J, Bongers P, Dieën J. Cumulative Low Back Load at Work as a Risk Factor of Low Back Pain: A Prospective Cohort Study. Journal of occupational rehabilitation. 2013;23(1):11-8.

7. Sparto PJ, Parnianpour M. Estimation of trunk muscle forces and spinal loads during fatiguing repetitive trunk exertions. Spine. 1998;23(23):2563-73.

8. Matsunaga M. Familywise Error in Multiple Comparisons: Disentangling a Knot through a Critique of O'Keefe's Arguments against Alpha Adjustment. Communication Methods and Measures. 2007;1:243-65.

9. Rubin M. Do p Values Lose Their Meaning in Exploratory Analyses? It Depends How You Define the Familywise Error Rate. Review of General Psychology. 2017;21:269-75.

---

## [Decision Letter · Decision Letter 1]

23 Jun 2021

Exploring lumbar and lower limb kinematics and kinetics for evidence that lifting technique is associated with LBP.

PONE-D-21-06065R1

Dear Dr. Saraceni,

We’re pleased to inform you that your manuscript has been judged scientifically suitable for publication and will be formally accepted for publication once it meets all outstanding technical requirements.

**Please, when revising the proofs, delete the word "gold-standard" in the abstract as this is not strictly true and may confound future readers.**

Kind regards,

Daniel Boullosa

Academic Editor

PLOS ONE

Additional Editor Comments (optional):

Reviewers' comments:

Reviewer's Responses to Questions

**Comments to the Author**

1. If the authors have adequately addressed your comments raised in a previous round of review and you feel that this manuscript is now acceptable for publication, you may indicate that here to bypass the “Comments to the Author” section, enter your conflict of interest statement in the “Confidential to Editor” section, and submit your "Accept" recommendation.

Reviewer #1: All comments have been addressed

Reviewer #2: All comments have been addressed

2. Is the manuscript technically sound, and do the data support the conclusions?

Reviewer #1: (No Response)

Reviewer #2: Yes

3. Has the statistical analysis been performed appropriately and rigorously? 

Reviewer #1: (No Response)

Reviewer #2: Yes

4. Have the authors made all data underlying the findings in their manuscript fully available?

Reviewer #1: (No Response)

Reviewer #2: (No Response)

5. Is the manuscript presented in an intelligible fashion and written in standard English?

Reviewer #1: (No Response)

Reviewer #2: Yes

6. Review Comments to the Author

Reviewer #1: (No Response)

Reviewer #2: (No Response)

7. PLOS authors have the option to publish the peer review history of their article (what does this mean?). If published, this will include your full peer review and any attached files.

Reviewer #1: No

Reviewer #2: No

---

## [Editor Report · Acceptance letter]

25 Jun 2021

PONE-D-21-06065R1 

Exploring lumbar and lower limb kinematics and kinetics for evidence that lifting technique is associated with LBP. 

Dear Dr. Saraceni:

I'm pleased to inform you that your manuscript has been deemed suitable for publication in PLOS ONE. Congratulations! Your manuscript is now with our production department. 

Kind regards, 

on behalf of

Dr. Daniel Boullosa 

Academic Editor

PLOS ONE